# Caesium-133 Accumulation by Freshwater Macrophytes: Partitioning of Translocated Ions and Enzyme Activity in Plants and Microorganisms

**Laura Zorza** [1,2], **Maris Bertins** [3], **Kristine Saleniece** [2,4], **Gunta Kizane** [2], **Andrejs Grinbergs** [1,2], **Ugis Eismonts** [2,4], **Ingars Reinholds** [2,3,*], **Arturs Viksna** [3] **and Olga Muter** [1,2]

1    Institute of Microbiology and Biotechnology, University of Latvia, 1 Jelgavas Str., 1004 Riga, Latvia; laura.zorza@lu.lv (L.Z.); andrejsg2@inbox.lv (A.G.); olga.mutere@lu.lv (O.M.)

2    Institute of Chemical Physics, University of Latvia, 1 Jelgavas Str., 1004 Riga, Latvia; kristine.saleniece@lu.lv (K.S.); gunta.kizane@lu.lv (G.K.); ugis.eismonts@lu.lv (U.E.)

3    Faculty of Chemistry, University of Latvia, 1 Jelgavas Str., 1004 Riga, Latvia; maris.bertins@lu.lv (M.B.); arturs.viksna@lu.lv (A.V.)

4    Faculty of Medicine, University of Latvia, 1 Jelgavas Str., 1004 Riga, Latvia

*    Correspondence: ingars.reinholds@lu.lv; Tel.: +371-2680-2448

**Abstract:** The potential of aquatic plants to accumulate Cs may be of notable importance in the environmental monitoring of radioactive wastes. This study aimed to evaluate the accumulation of Cs-133 by freshwater macrophytes *Bacopa amplexicaulis*, *Elodea densa*, *Ceratophyllum submersum*, and *Limnobium laevigantum* after a 10-day incubation period with CsCl (1–1000 μM). The partitioning of Cs and other elements, including 21 metals, such as P, B, and As, was analyzed using inductively coupled plasma mass spectrometry combined with principal component analysis (PCA). The enzymatic activity of plant crude extracts and aquatic microorganisms was characterized. The transfer factor (TF) reached the highest values of 0.13 and 0.10 for *C. submersum* and *L. laevigantum*, respectively, at 1000 μM Cs. The TFs in the other sets were below 0.1. In the presence of Cs-133, there was a significant increase in dehydrogenase activity ($p < 0.05$) and a decrease in the activity of the Folin–Ciocalteu assay. A three-fold decrease in culturable microorganisms was found in plants with 1000 μM Cs. PCA analysis revealed the species-specific elemental distribution in plant biomass and the aquatic phase. A negative correlation between Na, Ca (2.0–2.5, PC1) and Mg, K, and P (−2, PC1) was found. Certain enzyme groups can serve as bioindicators of Cs pollution in aquatic ecosystems.

**Keywords:** Cs-133; element analysis; antioxidant activity; enzymes; ICP-MS; macrophytes; transfer factor

## 1. Introduction

The growing global demand for nuclear energy poses the risk of the accidental release of nuclear products into the environment. According to data published by the European Nuclear Society in 2021, Europe has 176 nuclear power plants in operation, raising concerns about potential nuclear waste in the coming decade. Recently, a large amount of data on this topic was reported by Japanese researchers due to their experience with nuclear incidents affecting the release of Cs radionuclides into the environment. The Fukushima Daiichi Nuclear Power Plant disaster in 2011 resulted in radioactive caesium adsorption by soil and plants as a result of the transfer of radionuclides to livestock [1–5]. Other sites of nuclear mishaps, e.g., Ignalina [6] and Chernobyl, have been reported [7,8]. The accumulation risks of the disposal of radioactive wastes can also be attributed to Latvia and other post-Soviet countries due to the presence of old nuclear power plant facilities and their nuclear wastes located in these countries' territory [9].

In this respect, the health and ecological effects of radionuclide contamination are of great interest to various fields of research, including biomedicine, ecology, and biotechnology techniques for the removal of toxic metal compounds. Among radionuclides, radiocaesium (Cs-137, Cs-134) raises the most concern due to the relatively long half-life of Cs-137 (> 30 years) and the high degree of mobility and bioavailability of $Cs^+$ ions [10,11]. Anoxic remobilization of Cs from sediments and the release of Cs by macrophytes should be considered in order to understand and predict Cs distribution in the water column [12,13].

The partitioning of Cs among the abiotic and biotic compounds of aquatic ecosystems depends on many factors, e.g., mineralogical composition of suspended and bottom sediments, abundance of monovalent cations, and the characteristic geochemistry of water [10,11]. Many studies have been conducted with stable Cs-133 due to the similarities in the distribution of radiocaesium and Cs-133 in soil, sediment, and water [3,13–15].

The present study particularly focused on Cs distribution in aquatic environments and accumulation by macrophytes for the development of water treatment technologies, including wetlands, ponds, hydroponics, and other approaches [13,16]. In 1993, the overall accumulation of Cs-137 by macrophytes in a small, drainless lake in Belarus constituted 5% of its stock in the lake's water mass [8]. The quantification of Cs contamination in the field experiment showed that the contributions of the suspended particle fraction and aquatic biota were 80 and 20%, respectively [17]. Studies on Cs accumulation by plants have demonstrated a broad heterogeneity in results depending on species characteristics and environmental factors [1,3,13,18].

Moreover, the presence and accumulation of other metals in plants may have a significant impact on plant properties and indicate environmental contamination, which could help determine technological solutions. For example, in their experiments with water hyacinth (*Eichhornia crassipes*), Du et al. [19] found that, despite a good Me absorption capacity, the concentration of metals in leaves was low due to a low translocation factor. Therefore, after the remediation stage, the leaves of water hyacinth can be used as animal feed [19].

In this study, the accumulation of Cs by freshwater macrophytes was evaluated, in addition to concentrations of other elements in plant biomass. The processes of metal accumulation and remobilization in plants are supposed to be species-specific and distinct depending on the initial Cs concentration in water. An emphasis on extra-low concentrations of Cs (i.e., 1 μM) in water will bring this study closer to real water contamination with micropollutants, particularly radionuclides. The macrophytes *Bacopa amplexicaulis*, *Elodea densa*, *Ceratophyllum submersum*, and *Limnobium laevigantum* were selected for the study due to their reported potential to accumulate different metals in considerable quantities, whereas no data on Cs accumulation thus far have been reported in these studies [20–27]. Thus, it was critical to evaluate, within laboratory testing, whether these plans could be used to determine Cs accumulation.

The aim of this study was to analyse the accumulation of non-radioactive Cs (Cs-133) in freshwater macrophytes and to evaluate the potential toxic impact of caesium on plant species. It was hypothesized that this knowledge would have a necessary impact on a more detailed evaluation of the effect of radiocaesium (Cs-137, Cs-134) on the environment, including the accumulation in plant species, to exclude or confirm the summary effect of chemical toxicity and the impact on radiation. For this reason, the main evaluation in this study was provided solely by testing Cs-133 at different concentrations.

The experiment was performed in self-sustaining aquarium water for 10 days, with added initial caesium chloride (CsCl) concentrations ranging from 1 to 1000 μM. Plant responses to Cs-133 accumulation was evaluated by a range of enzyme groups, which are involved in antioxidant activities and metabolism, and the effect of multivariate factors, such as elemental contents and enzyme activity, was compared by statistical analysis.

## 2. Materials and Methods

### 2.1. Plants, Chemicals, and Reagents

Representative freshwater aquatic higher plants, namely, *B. amplexicaulis*, *E. densa*, *C. submersum,* and *L. laevigantum,* were purchased from local professional suppliers in Latvia. All the chemicals purchased for testing were of analytical grade and were used as received without any further purification. Ascorbic acid, CsCl, fluorescein diacetate (FDA), 3-methyl-2-benzothiazoninone hydrazine (MBTH), TRIS-HCl, 4-methylcatehol, polyvinylpyrrolidone (PVP), and Standard Count Agar (SCA) were purchased from Sigma-Aldrich (St. Lois, MO, USA). Guaiacol was purchased from Scharlau (Barcelona, Spain). Iodonitrophenyl formazan (INTF) was purchased from Fluka Analytical (Vienna, Austria). $Na_2HPO_4 \times 12H_2O$ and $Na_2CO_3$ were purchased from Lachema (Brno, Czech Republic). $KH_2PO_4$, acetone, Bradford reagent, 35% $H_2O_2$, Folin-Ciocalteau reagent were purchased from Enola (Riga, Latvia). L-3,4-dihydroxyphenylalanine dopaquinone (L-DOPA) was purchased from Alfa Aesar (Lancashire, UK).

### 2.2. Experimental Setup

The four plant species were cultured in a 56 L freshwater aquarium under sustainable conditions for six months. Then, each of the plant species was submerged in a 120 mL polypropylene flask containing 100 ml of aquarium water amended with 0, 1, 10, 500, and 1000 μM CsCl water solution. Samples were taken in triplicate to evaluate the range of element contents and provide microbiological and biochemical testing. The amount of plant biomass per flask was approximately 10% of the liquid volume. The experimental flasks were randomly placed and incubated at 23 °C for 10 days with a 12:12-h dark–light cycle and periodic shaking. Additional control sets were prepared with the same concentrations of CsCl added to the water samples without the plants. At the end of the incubation period, the volume of the aquatic phase was justified to the initial level by adding deionized water due to evaporation. Afterwards, the liquid phase and plant were separated and kept frozen until testing.

### 2.3. Physicochemical Characteristics of the Aquatic Phase

The water used in the study was considered to be self-sustaining aquarium water, derived from the freshwater aquarium, which had a sustained ecosystem (plants, fish, snails, shrimps, gravel with biofilm) for at least three months. The water pH was 6.8, conductivity was 563 μS/cm, concentrations of sodium, potassium, calcium, phosphorous, and magnesium in the freshwater were 28.97 mg/L, 4.61 mg/L, 3.64 mg/L, 74.46 μg/L, and 18.98 mg/L, respectively. Concentrations of nitrites and nitrates were below the level of detection (LOD) (data not shown).

### 2.4. Element Analysis

Before analysis, the moisture content was determined in plant species by drying partial samples at 105 °C until a constant weight, according to laboratory practices of plant nutritional analysis [28].

The analysis of elemental composition (e.g., 21 metals, B, P, and As) was conducted using an Agilent 8900 (USA) inductively coupled plasma triple quadrupole mass spectrometry (ICP-QQQ-MS) device equipped with a MicroMist nebulizer. The results of determined element concentrations, except Cs, are summarized in Figure S4 of the Supplementary Material. The instrumental parameters of ICP-MS were set as follows: RF power —1.550 W; auxiliary gas flow—0.90 mL/min; plasma gas flow—15.0 L/min; He cell gas flow—5.0 mL/min; and sampling depth—8 mm. The analytical standard stock solutions were purchased from TraceCert®(Sigma-Aldrich, Burlington, MA, USA) for ICP (100 mg/L). The instrumental limit of detection (LOD) for the determined elements was 0.02 μg/L and the limit of the quantification (LOQ) was 0.06 μg/L. The linear range for the measurements by the ICP-MS method was up to 10 mg/L. The relative standard deviation (RSD) within the analysis of one sample replicate did not exceed 6%. For the calibration

graph, standard solutions in the concentration ranged from 0.1 to 100.0 μg/L. Calibration solutions were prepared from stock standard solutions. A calibration graph with blank correction was used to calculate the concentrations of elements in the samples. An internal standard mix solution from Agilent Technologies (10 g/mL) was used as an internal standard. A stability check of the ICP-MS system was performed by using two standard solutions after every 10 samples. For quality control of the analytical procedures, the same elements were also determined using reference material IAEA-336 provided by the International Atomic Energy (Vienna, Austria) containing 45 elements.

The reference material sample was included at the beginning and end of each analytical sequence. The differences between the certified and measured values did not exceed 10%. Three replicates of each sample were used for the analysis. A Mass Hunter workstation programme, together with its Instrument control and Offline data analysis programmes, was used.

The transfer factor (TF) of Cs-133 from the aquatic phase to plants was calculated according to [29], with minor modifications, as stated in Equation (1):

$$TF = \frac{(final\ metal\ concentration\ in\ plant) - (initial\ metal\ concentration\ in\ plant)}{metal\ concentration\ in\ aquatic\ phase} \tag{1}$$

### 2.5. Microbiological Analysis

The number of colony-forming units (CFU) was determined by the microdilution plating method. Sample dilutions ($10^{-1}$ to $10^{-6}$) were prepared for each sample in microplates. Using a multichannel pipette, 10 μL of each sample dilution was used to inoculate Petri dishes containing SCA medium. The plates were incubated at 30 °C for 48 h. The data obtained were processed in triplicate and determined as CFU/mL.

### 2.6. Testing of Enzyme Activity in the Aquatic Phase and Crude Plant Extracts

#### 2.6.1. Aquatic Phase

The enzyme activity of microorganisms in water was measured by the FDA hydrolysis assay according to the procedure reported, with some modifications [30]. The water samples were transferred to a 12-well microplate (600 μL per well). After the addition of 2.4 mL FDA reaction mixture (4 mg FDA, 2 mL acetone, 48 mL 60 mM phosphate buffer) to each well, the plate was incubated for 48 h at 37 °C. Afterwards, the reaction was stopped by the addition of 600 μL acetone. The samples were transferred to 1.5 mL reaction microtubes and centrifuged at 10,000 rpm for 5 min. The supernatant was transferred to a 96-well microplate (200 μL per well). The hydrolysed FDA concentration was determined photometrically using a TECAN Infinite F50 microplate reader at OD492 (Männedorf, Switzerland).

#### 2.6.2. Plant Crude Extract

#### Preparation of the Plant Crude Extract

Frozen plant samples were washed twice with distilled water. Ten grams of the plant sample was cut into thin slices and homogenized in buffer medium (50 mL 100 mM sodium phosphate buffer (pH 7.0) containing 1 mM ascorbic acid and 0.5% (*w/v*) PVP) for 5 min at 4 °C. The homogenate was filtered through three layers of cheesecloth. The filtrate was then centrifuged at 5000× *g* for 15 min, and the supernatant was collected according to the procedure reported by [31].

#### Protein Concentration

The concentration of proteins in crude plant extracts was determined using the Bradford assay. The measurement was performed in 96-well microplates with 200 μL Bradford [32] reagent and 100 μL of sample per well. The calibration curve was prepared using bovine albumin with good linearity ($R^2 = 0.96$). Before testing the enzyme activity, plant crude extracts were justified by protein concentrations.

DOPA-MBTH Assay

The assay measures the pink pigment formed by the reaction of Besthorn's hydrazone (MBTH) with dopaquinone, the product of oxidation of L-DOPA by tyrosinase [33]. The reaction mixture of 45 mg L-DOPA and 10 mg MBTH was dissolved in 25 mL 50 mM phosphate buffer at pH 6.0, and 500 µL crude extract in a total volume of 3.0 mL was prepared at 23 °C. The increase in absorption was measured photometrically at 492 nm. The assay was carried out in triplicate.

Dehydrogenase Activity

The dehydrogenase (DHA) activity was determined by the reduction of 2-p-iodo-3-nitrophenyl-5-phenyltetrazolium chloride (INT) to iodo-nitrophenyl formazan (INTF), according to [34] with some modifications. Two mL of the crude plant extract sample was centrifuged at 10,000 rpm for five minutes. The supernatant was discarded. The reaction was performed on a concentrated sample by adding 200 µL of the reaction mixture (20 mL of 0.25 M TRIS-HCl, 40 mg INT, 10 mg glucose). The samples were incubated for 24 h at 28 °C in triplicate, after which the absorption was measured at 492 nm using a Tecan Infinite F50 microplate reader (Switzerland).

Peroxidase Activity

The peroxidase activity was determined using 4-methylcatechol as the substrate. The reaction mixture (100 mM sodium phosphate buffer (pH 7.0), 5 mM 4-methylcatechol, 5 mM $H_2O_2$, and 500 µL of crude extract in a total volume of 3.0 mL) was prepared at 23 °C. The increase in the absorption caused by oxidation of 4-methylcatechol by $H_2O_2$ was measured photometrically at 420 nm [35]. The assay was carried out in triplicate.

The Folin–Ciocalteu Assay

The Folin–Ciocalteu (F-C) reaction is an antioxidant assay based on electron transfer, which measures the reductive capacity of an antioxidant. A plant extract aliquot was mixed with a diluted Folin–Ciocalteau reagent (2.5 mL, 10%) and $Na_2CO_3$ (4%, 2.0 mL) aqueous solution, as described by [36]. After 24 h in the dark, absorbance was measured at 740 nm using a spectrophotometer. A control sample was performed with 500 µL distilled water, 2.5 mL Folin–Ciocalteau reagent, and 2 mL $Na_2CO_3$ solution. The assay was carried out in triplicate.

Guaiacol Peroxidase Activity

Guaiacol peroxidase is involved in many important biosynthetic processes and defence against stress, e.g., exposure to polycyclic aromatic hydrocarbons, heavy metals, and herbicides [37]. The reaction mixture contained 80 mM guaiacol and 10 mM $H_2O_2$ in 0.066 M K-Na phosphate buffer (pH = 7.4), with the addition of polyvinylpyrrolidone according to [38]. The enzymatic reaction was started by adding 0.1 mL of the crude plant extract to 3 mL of the reaction mixture. The activity of guaiacol peroxidase was measured photometrically at 492 nm. The assay was carried out in triplicate.

*2.7. Statistical Analysis*

A one-way analysis of variance (ANOVA) was applied to evaluate the differences between the obtained results. The level of significance was set to $p < 0.05$. All statistical analyses were performed using Microsoft Excel and CAT (Chemometrics agile tool) software (Gruppo di Chemiometria della Divisione di Chimica Analitica della Società Chimica Italiana).

For multivariate statistical analysis, Statistica for Windows 6.0 software packages (StatSoft, Tulsa, OK, USA) were used. The calculation of mean concentrations of macro- and trace elements in plants and water samples and the analysis of variance to estimate statistically significant differences between groups of samples were carried out. To estimate the relationships between element concentrations in plants, Pearson's correlation coeffi-

cients were calculated. Additionally, principal component analysis (PCA) was applied to the experimental data sets. The data for PCA were normalized to unit concentration to avoid misclassifications caused by different orders of magnitude in the variables. The combination of PCA can assess the contribution of specific factors that may affect the variations in the elements in different plant parts and appreciate the mechanisms influencing the element distribution in each plant species regarding the effect of CsCl concentration.

## 3. Results

### 3.1. Accumulation of Cs and Other Metals by Aquatic Plants

The accumulation of Cs by macrophytes was measured after 10 days of incubation. As shown in Figure 1A, the initial Cs concentration greatly influenced the intensity of Cs accumulation in the plant biomass. Thus, the amount of Cs in biomass in the sets with 1000 μM Cs was 1.5–2.0 orders higher than that with 10 μM Cs. This trend was shown for all tested plants. At the lowest Cs concentration (i.e., 1 μM), its accumulation in plants was negligible, particularly 29.1, 35.3, 22.0, and 28.1 mg/kg dw for *B. amplexicaulis*, *E. densa*, *C. submersum,* and *L. laevigantum*, respectively (Figure 1A). One of the most important criteria for evaluating the metal-accumulation potential of plants is the transfer factor (TF) of Cs-133 in the plants. The data on Cs TF values for the macrophytes under the tested conditions are summarized in Table 1. The highest TF values were detected for *C. submersum* and *L. laevigantum* at 1000 μM Cs, which reached 0.13 and 0.10, respectively. Other measurements of TF for different initial Cs concentrations and four tested plants were below 0.1 (Table 1).

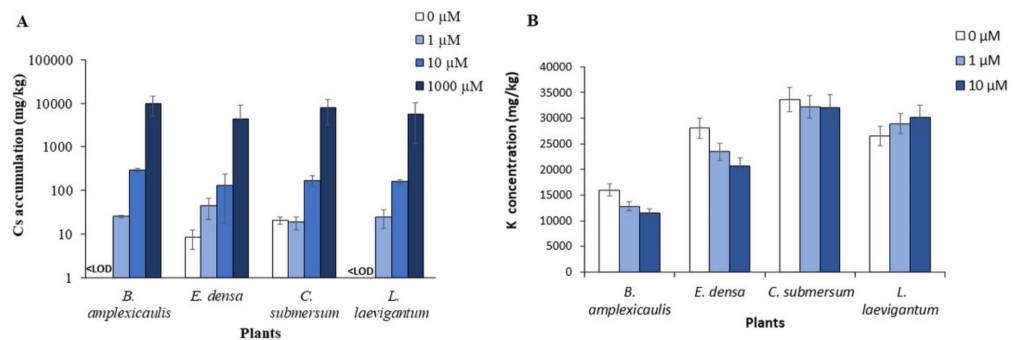

**Figure 1.** Accumulation of Cs (**A**) and K (**B**) in aquatic plants after incubation with different concentrations of CsCl for 10 days. <LOD values are below the LOD limit of detection.

**Table 1.** The transfer factor of Cs-133 after the 10-day incubation of macrophytes in the aquatic phase with different Cs concentrations.

| Initial Cs Concentration (μM) | Plants | | | |
|---|---|---|---|---|
| | *B. amplexicaulis* | *E. densa* | *C. submersum* | *L. laevigantum* |
| 1 | 0.09 | 0.00 | 0.05 | 0.06 |
| 10 | 0.09 | 0.05 | 0.09 | 0.04 |
| 1000 | 0.02 | 0.04 | 0.13 | 0.10 |

Our data on Cs and K concentrations in plant biomass have been tested for their possible correlation depending on the initial Cs concentrations. Putting together all the data related to 0, 1, 10, and 1000 μM Cs, the correlation analysis showed a strong species-specific effect. In particular, the concentrations of Cs and K in plant biomass were positively correlated in *B. amplexicaulis* ($R^2 = 0.61$) and *E. densa* ($R^2 = 0.50$), while not correlated in *C. submersum* ($R^2 = -0.00$) and negatively correlated in *L. laevigantum* ($R^2 = -0.79$) (Figure S1).

### 3.2. Microbial Abundance and Enzyme Activity in an Aquatic Phase Depending on the Cs Concentration

As was reported earlier, direct biological Cs accumulation from water/sediment/soil occurs readily due to primary producers (i.e., plants) and microorganisms [10,11]. At the beginning of the experiment, an aquatic phase contained a sustained microbial consortium. Therefore, further incubation of different plants with Cs was expected to change the microbial activity. The FDA hydrolysis activity assay was applied for estimation of microbial response towards Cs-133 in an aquatic phase during the 10-day experiment. As shown in Figure 2A, FDA hydrolysis activity did not show any trend with increasing Cs concentration in the aquatic phase. No significant plant species-specific differences were revealed. Another parameter that could indicate the effect of Cs on the microbial community is the number of culturable microorganisms. As shown in Figure 2B, a decrease in the CFU count by three orders in the presence of 1000 μM Cs was found compared to the control set without Cs. The exceptions were the sets with *E. densa* and non-planted sets, where the inhibition of culturable microorganisms by 1000 μM Cs was negligible (Figure 2B).

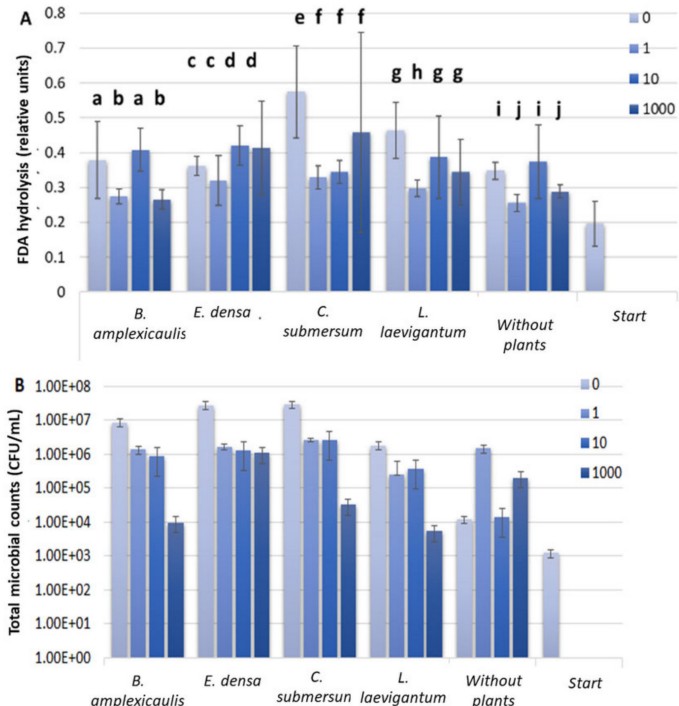

**Figure 2.** Fluorescein diacetate (FDA) hydrolysis assay (**A**) and CFU counts (**B**) in an aquatic phase before and after plant incubation with different concentrations of CsCl for 10 days. Concentrations of CsCl: 0, 1, 10, and 1000 μM. The FDA hydrolysis assays were incubated for 72 h. For variables with different letters, the difference between the means is statistically significant ($p < 0.05$). The differences are shown within the dataset for one plant individually.

### 3.3. Changes in Enzyme Activity in Plant Crude Extract after Plant Incubation with Cs

In our study, the physiological response of macrophytes to Cs-133 was assessed by the changes of different groups of enzymes, i.e., DHA, which reflects the electron transport system activity, correlated with the respiration rate [39]; peroxidase and guaiacol peroxidase–enzymatic antioxidants [40], L-DOPA+MBTH assay–phenol oxidases activity [41], and Folin–Ciocalteu–polyphenols–nonenzymatic antioxidants [40]. Some of these reactions developed relatively slowly and were thus measured after 24 h incubation (Figure 3), while others reacted immediately after the addition of the enzyme substrate and monitored for the first 15 min (Figure 4).

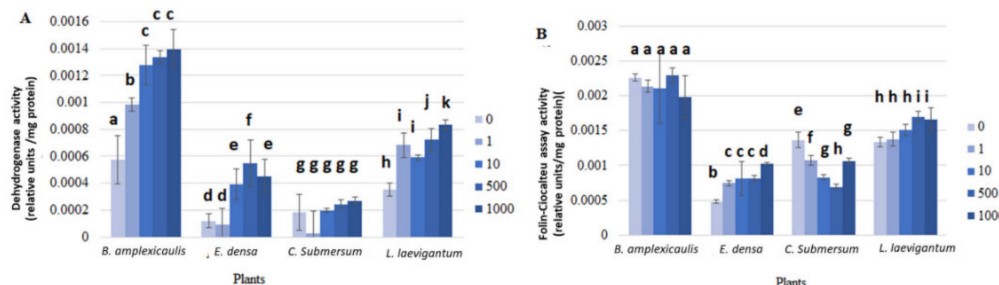

**Figure 3.** Dehydrogenase (**A**) and Folin–Ciocalteu (**B**) assays with crude plant extracts after plant incubation with different concentrations of CsCl for 10 days. Concentrations of CsCl: 0, 1, 10, and 1000 μM. A—Dehydrogenase; B—Folin–Ciocalteu assay. Enzyme activity was measured after 24 h of incubation. For variables with different letters, the difference between the means is statistically significant ($p < 0.05$). The differences are shown within the dataset for one plant individually.

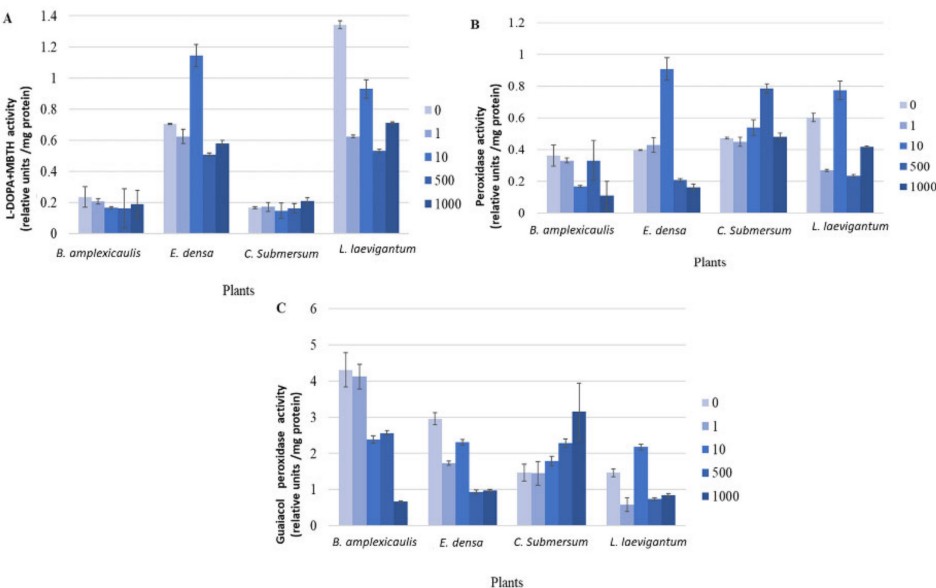

**Figure 4.** L-DOPA + MBTH (**A**), peroxidase (**B**), and guaiacol peroxidase (**C**) assays with crude plant extracts after plant incubation with different concentrations of CsCl for 10 days. Concentrations of CsCl: 0, 1, 10, and 1000 μM. Enzyme activity of crude plant extracts is presented in units defined as a 0.001 change in absorbance per minute.

The activity of DHA gradually increased in the crude plant extract with increasing Cs concentration in the aquatic phase. In particular, the enzyme activity in *B. amplexicaulis*, *E. densa*, *C. submersum,* and *L. laevigantum* after incubation at 1000 μM Cs was 142, 279, 48 and 137% higher, respectively, than in the control without Cs (Figure 3A). An increase of DHA activity in the presence of 1 μM Cs was significant ($p < 0.05$) in the sets with *Bacopa* sp. and *Limnobium* sp. compared with the control. A further significant ($p < 0.05$) increase in DHA activity with increasing Cs concentration from 1 to 10 μM Cs was observed for *B. amplexicaulis* and *E. densa* (Figure 3A).

The FC assay revealed a slight Cs activation effect for *E. densa* and *L. laevigantum,* while no effect and a gradual decrease in *B. amplexicaulis* ($p > 0.05$) and *C. submersum*, respectively, were detected (Figure 3B). Specifically, a gradual significant ($p < 0.05$) decrease in the activity was detected in *C. submersum* at 1, 10, and 500 μM Cs (Figure 3B).

The fast development of a colour reaction in the L-DOPA, peroxidase, and guaiacol peroxidase assays revealed the more precise differences in the kinetics of antioxidant response in four tested plants depending on the Cs concentration (Figures S2–S4 of the Supplementary Material). A considerable increase of enzyme activity at 10 μM Cs, compared with

the control, was shown in *E. densa* in DOPA and peroxidase assays and in *L. laevigantum* in a guaiacol peroxidase assay (Figures S2 and S3). More precisely, quantification of these differences (i.e., change in absorbance per minute) showed an increase in the mentioned enzymes 1.61, 2.28, and 1.49 times compared to the control, respectively (Figure 4).

### 3.4. Principal Component Analysis

Previous studies showed that in the case of a large amount of multivariate data, PCA can be effectively used to indicate covariation in different measured parameters between different plant species [42].

PCA of the studied plants after plant incubation with different concentrations of CsCl showed that the most variability we observed was in Component 1 (PC1, 41.4% of variance). Cs (−0.1 on PC1) is not the main factor that varies in different samples. The differences in PC1 are explained by the different elemental patterns of each plant species. There is good separation of different species on PC1. *C. submersum* forms a separate cluster with a centre at −3.4 on PC1, *L. laevigantum* and *E. densa* with a centre at 0 on PC1, and *B. amplexicaulis* with a centre at +3.2 on PC1. The main differences are related to the negative correlation of Na and Ca (1.8–2.6, PC1) and Mg, K, and P (−1.8–−2.6, PC1). Component 2 (PC2, 17.4%) shows the impact of increasing the Cs concentration on different plant species. There is practically no impact of increasing the Cs concentration on *L. laevigantum,* while there is a negative correlation between Cs and Mg content in *C. submersum* and a negative correlation between Cs and Mg content in *B. amplexicaulis*. Significant changes in the content of macroelements for *C. submersum*, *B. amplexicaulis*, and *E. densa* appear at a very high Cs content (1000 μM CsCl) (Figure 5). The accumulation of different elements in plant biomass after the 10-day incubation is shown in Figure S1 of the Supplementary Material. PCA also shows that the dehydrogenase and Folin–Ciocalteu assays covariate with Cs concentration and are most expressed in *B. amplexicaulis*. Guaiacol peroxidase increases with increased Cs concentration and is most expressed in *C. submersum* L-DOPA + MBTH, and peroxidase activity decreases with the increase in Cs concentration. These changes are more expressed in *E. densa* and *L. laevigantum*.

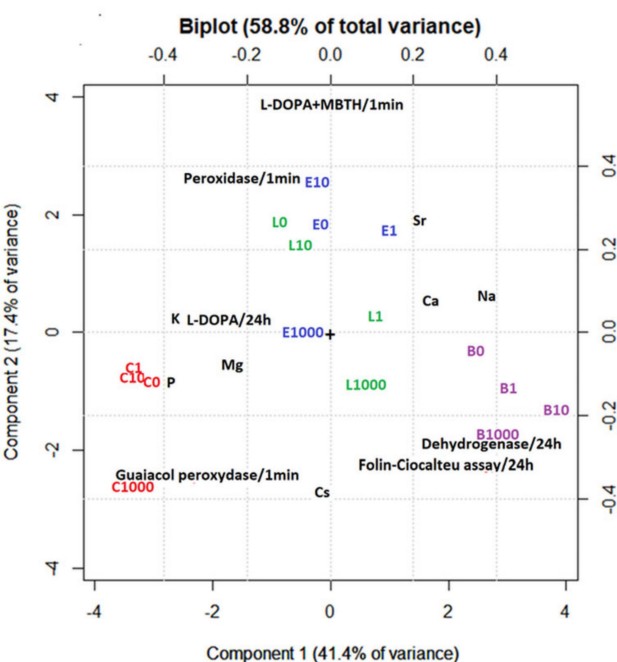

**Figure 5.** Principal component analysis (PCA) of macroelements in the studied plants (B—*B. amplexicaulis*, C—*C. submersum*, E—*E. densa,* and L—*L. laevigantum*) after plant incubation with different concentrations of CsCl for 10 days. Concentrations of CsCl: 0, 1, 10, and 1000 μM.



## 4. Discussion

Our results showed that the amount of Cs accumulated by macrophytes depended on the initial Cs concentrations in an aquatic phase. This observation is in good agreement with [14], who reported that concentrations of endogenous Cs were correlated with those in the environment, particularly sediments [14]. In this respect, the possible adsorption mechanisms of Cs-133 can explain the dependence of Cs accumulation on the initial concentration of Cs in water. This mechanism was proposed by [14] in model experiments with *E. densa* in hydroponics with predominant localization of Cs-133 in the cell wall or apoplectic regions. The same dependence was also reported to be attributed to other metals, e.g., Co [43]. However, the non-even distribution of Cs-133 in plant organs with greater amounts of Cs in younger parts of stems and leaf blades indicated the selective sorption of Cs by plants [4]. Vacula et al. [15] recently reported on Cs accumulation by *C. demersum* L. under laboratory and field conditions with a Cs concentration range comparable with that in our study. Thus, in the sets with initial Cs concentrations of 8 and 1067 μM after 8 days, the plant biomass accumulated on average 78 and 6248 mg/kg Cs, respectively [15]. In our study, after 10 days, at initial Cs concentrations of 10 and 1000 μM, 201.3 and 8763.2 mg/kg Cs, respectively, were accumulated by *C. submersum* (Figure 1A).

The possible mechanisms responsible for metal accumulation in submerged plants have been summarized by [44]. This review demonstrated a broad variability of the extent of metal accumulation in aquatic plants, depending on plant species, environmental conditions (including metal abundance and concentration), and experiment design. Among the protective functions of submerged plants against metal cytotoxicity, access control, rapid translocation, efficient accumulation, and detoxification are indicated [44]. The key roles in metal homeostasis and tolerance use such proteins as heavy metal-transporting ATPases, natural resistance-associated macrophage proteins, cation diffusion facilitator family proteins, and multidrug and toxin efflux proteins [45].

The values of TF are species-specific and greatly depend on the environmental conditions and the period of plant contact with metals [3,15]. For example, the TF of Cs from soil by shoots and brown rice ranged from 0.018 to 0.068 and 0.004 to 0.065, respectively [1]. Shinano et al. [3] compared the TF of radiocaesium (Cs-137, Cs-134) in 33 varieties of terrestrial *Amaranthus* spp. and indicated species-specific differences, which varied in the range from 0.048 to 0.170 [3]. Among the TF, other parameters are widely used in aquatic toxicology, e.g., bioconcentration and bioaccumulation factors. These factors are calculated by similar equations however differ by considering either only the dissolved phase or both the dissolved and particulate phases of a contaminant in the environment [46]. In addition, root and shoot accumulation factors, as well as translocation factors, are used for distinguishing metal localization in the plant [47].

Most of the studies focusing on Cs uptake by plants also emphasized Cs and K interrelations in the plants. In particular, the removal of Cs from water/sediments/soil often correlated with the K concentration. Absorption of $Cs^+$ into the roots was suggested to have occurred using two types of $K^+$ transporters (i.e., high affinity and low affinity) [48]. Our results indicate a strong species-specific effect. This fact is concordant with the studies of other authors [49]. In a study on Cs-137 and K-40 distribution in a lake, a negative correlation of Cs accumulation by *Phragmites australis* and K concentration was found [7]. Another field trial for 552 days in a pond reservoir spiked with Cs-133 demonstrated a correlation of Cs-133 removal with a period of increasing K concentration in the water, e.g., in winter [12]. As reported by [43], a high discrimination of the K+ transporter against Cs+ on the plant cell membranes takes place at enhanced K concentrations, i.e., above 0.3 mM. In our study, the K concentration in water did not exceed 0.1 mM (p.2.3), so it is unlikely to affect Cs accumulation in plants under the tested conditions. Inhibition of Cs uptake by plants can also occur in the presence of divalent ions, i.e., $Ca^{2+}$, $Mg^{2+}$, and $Ba^{2+}$ [50,51].

Another factor that could facilitate Cs removal from aquatic environments is microbial activity. Microorganisms can directly interact with and change the physicochemical properties of certain elements [44,52].

The response of microorganisms to Cs-133 in an aquatic phase in our study was evaluated by enzyme activity and CFU count. The FDA hydrolysis assay is based on the ability of bacteria or fungi to split the FDA molecule using several enzymes (e.g., esterases, lipases, proteases) [39]. FDA hydrolysis activity may correlate with the total physiological activity of heterotrophic microorganisms in environmental samples, which reflects local/microhabitat conditions [53]. Interestingly, FDA hydrolysis activity was not significantly changed at different Cs concentrations, while the number of CFU decreased by three orders of magnitude at 1000 μM CsCl. The mechanisms of interrelations "Cs: microorganisms" are still poorly understood. Nevertheless, some interesting data can be found for explaining the physiological processes occurring in microbial cells under Cs stress. Thus, in the study on yeast response to Cs and other monovalent cations, the yeast's ability to substitute for K in the activation of several enzymes was assessed [54]. Pyruvate kinase from *Rhodotorula rubra* was more inhibited by cations, whose ionic radii are smaller (e.g., $Na^+$ (97 pm) and $Li^+$ (78 pm)) or larger ($Cs^+$ (166 pm) compared to $K^+$ (133 pm)). At the same time, an abundance of Cs of up to 100 mM did not affect a pyruvate kinase from *Saccharomyces cerevisiae* [54]. These observations support the hypothesis on species specificity in microbial response to metal stress.

As previously reported in the discussion, the influx of Cs into the plant biomass occurs via $K^+$ transporters. Potassium regulates the osmotic pressure of cells, which in turn is essential for enzymatic reactions. In cases where the increased Cs concentrations replace K in cells, the cytotoxicity increases due to inhibition of enzymes [51,55,56]. The results obtained in this study indicated that the dependence of K concentration in the plant tissues on Cs accumulation was plant species-specific. In this respect, the changes in enzyme activity in the plant crude extract after incubation with Cs in different concentrations would bring new insight into plant ecotoxicological response towards Cs.

L-DOPA is known to be one of the most used substrates for phenol oxidase (PO) and peroxidase assays in soil and other environmental samples [41]. The effect of divalent metals (up to 800 mM) on PO activity in soil was tested by [57]. PO activities were stimulated by Mg(II), Mn(II), and Zn(II) and were inhibited by other metals [57]. Thus far, no data on the Cs effect on PO activity in aquatic plants under metal stress has been reported. Our data corroborate with other studies, which showed that Cs-133 affects plant functioning via activating the defence mechanism against oxidative stress [58,59].

In a recent study with a freshwater macro-algae *Nitella pseudoflabellata*, an increased activity of catalase and guaiacol peroxidase was observed at 0.75 μM Cs-133 compared to the control [58]. This effect is also attributable to other metals. Thus, an increasing activity of antioxidant enzymes has also been detected in macrophyte *Hydrilla verticillate* when exposed to cobalt (up to 100 μM) for seven days [43]. However, in the study with *Elodea* sp., activation of antioxidant enzymes was detected during the first eight hours of exposure to nickel, which corresponds to the passive binding of metal ions (chemisorption) [60].

Nevertheless, the metal-induced activation of antioxidant enzymes in plants is determined by species-specific characteristics. In the comparative study on Cd accumulation by the semi-aquatic plant *Nymphoides peltata* and the aquatic plant *Nymphaea* sp., a similar level of Cd accumulation was shown; however, the plant resistance towards Cd differed. In particular, Nymphaea had greater peroxidase activity and was more resistant to Cd than *N. peltata* [61]. In our study, the highest concentration of Cs, i.e., 1000 μM, resulted in the strongest inhibition of peroxidase and guaiacol peroxidase activity in *B. amplexicaulis*. However, no linear dependence of enzyme activity on Cs concentration was observed.

## 5. Conclusions

The accumulation of caesium in macrophytes was greatly influenced by an initial concentration added during the aquatic phase. The amount of Cs in biomass in the sets with

1000 µM Cs was 1.5–2.0 orders higher than that with 10 µM Cs. This trend was indicated for all tested plants. The PCA analysis revealed a species-specific elemental distribution in plant biomass and the aquatic phase. The main differences could be related to the negative correlation of Na and Ca (2.0–2.5, PC1) and Mg, K, and P ($-2$, PC1).

An inhibitory effect of 1000 µM Cs on aquatic microorganisms was detected; specifically, the CFU count was decreased by up to three orders of magnitude, as compared to the set without Cs. Nevertheless, FDA hydrolysis activity did not show any considerable changes after incubation with different Cs concentrations.

The activity of DHA gradually increased in the crude plant extract with increasing Cs concentration in the aquatic phase. Importantly, this effect was also detected at 1 µM Cs in the sets with *B. amplexicaulis* and *L. laevigantum* compared with the control ($p < 0.05$).

A considerable increase in antioxidative enzyme activity at 10 uM Cs, compared with the control, was shown in *E. densa* in DOPA+MBTH (1.61 times) and peroxidase assays (2.28 times) and in *L. laevigantum* in a guaiacol peroxidase assay (1.49 times) compared with the control.

This study has revealed certain enzyme groups in the tested aquatic plants that specifically respond to the presence of Cs in water, e.g., dehydrogenase and guaiacol peroxidase. These parameters could serve as bioindicators for the ecotoxicological evaluation of Cs-contaminated sites. Further intensive research is needed to describe the mechanisms of Cs accumulation in aquatic plants in environmental, technological, and ecotoxicological aspects. These results provide important information about the effect of Cs ions and can also be attributed to the evaluation of radioactive Cs accumulation in plants. It can be concluded that there is a low effect of Cs-133 on plant species at low concentrations; thus, in the case of Cs-137, radioactivity may be the main factor affecting plant species and should be addressed in further studies.

**Supplementary Materials:** The following are available online at https://www.mdpi.com/article/10.3390/su14031132/s1, Figure S1. Accumulation of different elements in *B. amplexicaulis*, *E. densa*, *C. submersum*, and *L. laevigantum*. Biomass after 10-day incubation in the presence of 0, 1, 10, and 1000 µM CsCl. Figure S2. Dopaquinone and 3-methyl-2-benzothiazoninone hydrazine (DOPA+MBTH) activity of crude plant extracts (A—*B. amplexicaulis*, B—*E. densa*, C—*C. submersum*, D—*L. laevigantum*) after plant incubation with 0 to 1000 µmol/L (µM) CsCl for 10 days. Figure S3. Peroxidase activity of crude plant extracts (A—*B. amplexicaulis*, B—*E. densa*, C—*C. submersum*, D—*L. laevigantum*) after plant incubation with 0 to 1000 µmol/L (µM) CsCl for 10 days. Figure S4. Guaiacol peroxidase activity of crude plant extracts (A—*B. amplexicaulis*, B—*E. densa*, C—*C. submersum*, D—*L. laevigantum*) after plant incubation with 0 to 1000 µmol/L (µM) of CsCl for 10 days.

**Author Contributions:** Conceptualization, O.M., A.G., A.V. and I.R.; methodology, O.M., I.R. and M.B.; software, I.R. and M.B.; validation, A.V., A.G. and M.B.; formal analysis, L.Z., K.S. and U.E.; investigation, L.Z., O.M., I.R., A.G., A.V. and M.B.; resources, G.K. and M.B.; data curation, A.V. and A.G.; writing—original draft preparation, L.Z., O.M., M.B. and I.R.; writing—review and editing, I.R., O.M., L.Z., M.B., A.G. and A.V.; visualization, O.M., I.R. and M.B.; supervision, A.G., A.V. and I.R.; project administration, I.R. and G.K.; funding acquisition, G.K. and I.R. All authors have read and agreed to the published version of the manuscript.

**Funding:** This research was funded by the Latvian Council of Science: Project No. lzp-2020/2-0213 "Application of modified cellulose fiber sorbents for concentration of radioactive hydrogen (tritium) and other radionuclides from contaminated environments and evaluation of toxic properties: approbation of the method for purification of water residues from nuclear reactors stored in Latvia."

**Institutional Review Board Statement:** Not applicable.

**Informed Consent Statement:** Not applicable.

**Data Availability Statement:** Not applicable.

**Acknowledgments:** The Latvian Council of Science is acknowledged for supporting this study and the further research based on these results. The fluent English speaker, M. Semjonova-Reinholde and the MDPI English editor provided proofreading.

**Conflicts of Interest:** The authors declare no conflict of interest.

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
