# Peer review of "Caesium-133 Accumulation by Freshwater Macrophytes: Partitioning of Translocated Ions and Enzyme Activity in Plants and Microorganisms"

_sustainability, doi:10.3390/su14031132_

Round 1
Reviewer 1 Report
The manuscript is interesting and in the scope of the journal. Correctly written with the possibility of the treatment application. Describes the accumulation of Cesium-133 by freshwater macrophytes, partitioning of translocated ions and enzyme activity in plants and microorganisms. The results illustrate decontamination approach has been used very effectively to improve the reliability of the wide application for this research.
Author Response
Authors are grateful for valuable comments.
All the corrections are presented in the manuscript by track changes.
There are the comments to the Reviewer's 1 valuable comments:
- Extensive editing of English language and style required. Response: Thank you for the reminder, the revised manuscript has been proofread by English speaking native.
- For empirical research, are the results clearly presented? Can be improved. Response: Thank you for this comment. The Figure 1B has been revised, while Figures 2A and 2B were formatted.
Reviewer 2 Report
Comments:
In this work, the authors carried out the study (in the laboratory) of the accumulation of Cs-133 in widely abundant freshwater macrophytes Bacopa amplexicaulis, Elodea densa, Ceratophyllum submersum and Limnobium laevigantum after the accumulation of certain concentrations of CsCl (1- 1000 µM). The work is interesting and in line with the scope of journal. The reviewer finds the article technically well written and organized. The hypothesis is valid and correct and based on solid theoretical and scientific background. In this context, the reviewer suggests minor changes in order to improve the presentation of the paper.
The critical comments are as follows:
Abstract:
Although present abstract highlights the results, however, it does not give any brief account of the background to deduce the justification for this work. Normally an abstract should include the above along with stating briefly the purpose of the study undertaken and meaningful conclusions based on the obtained results. Hence, this needs rewriting. I would expect brief, yet concise, the quantitative description in the abstract.
Introduction
The introduction should be improved, in order to clearly state the problem and establish a clear objective according to the title proposed for the manuscript. It is suggested:
It is recommended to review and cite the following references:
H.B. Ortiz-Oliveros, D. Cruz, R.M. Flores-Espinosa, I. Santillán-Malaquías, G. Zarazúa-Ortega, A. Villalva, Evaluation of the bioaccumulation of heavy metals and 137Cs in succulent plants Echeveria elegans, Int. J. Environ. Sci. Technol. (2021). https://doi.org/10.1007/s13762-021-03186-5.
Du Y, Wu Q, Kong D, Shi Y, Huang X, Luo D, Chen Z, Xiao T, Leung JYS (2020) Accumulation and translocation of heavy metals in water hyacinth: Maximising the use of green resources to remediate sites impacted by e-waste recycling activities. Ecol Indic 115:106384. https://doi.org/https://doi.org/10.1016/j.ecolind.2020.106384
Geng N, Wu Y, Zhang M, Tsang DCW, Rinklebe J, Xia Y, Lu D, Zhu L, Palansooriya KN, Kim KH, Ok YS (2019) Bioaccumulation of potentially toxic elements by submerged plants and biofilms: A critical review. Environ Int 131:105015. https://doi.org/10.1016/j.envint.2019.105015
Hao Z, Chen L, Wang C, Zou X, Zheng F, Feng W, Zhang D, Peng L (2019) Heavy metal distribution and bioaccumulation ability in marine organisms from coastal regions of Hainan and Zhoushan, China. Chemosphere 226:340–350.
Rai H, Kawabata M (2020) The dynamics of radio-cesium in soils and mechanism of cesium uptake into higher plants: Newly elucidated mechanism of cesium uptake into rice plants. Front. Plant Sci. 11:528
- It is necessary to clearly define the objective based on the proposed title. The objective is not clear, and the novelty of the study is not evident
Methodology:
- It is necessary to mention the purity of reagents, number of analyzes, uncertainty, etc.
- Why do the authors only calculate TF?
Discussions
- Development of the results and discussions is required to be in accordance with the proposed methodology.
- The discussion section is superficial; a more thorough discussion is required.
In reviewer opinion, these sections should be rewritten.
Conclusion:
- The conclusion should be rewritten based on the review of the discussions.
- Conclusion section is superficial. What are authors own viewpoints? What are the major findings and how they are addressing the left behind research gaps and current challenges?
Author Response
Authors are grateful for valuable comments.
All the corrections are provided in the manuscript by Track Changes function
Reviewer #2:
- Abstract: Although present abstract highlights the results, however, it does not give any brief account of the background to deduce the justification for this work. Normally an abstract should include the above along with stating briefly the purpose of the study undertaken and meaningful conclusions based on the obtained results. Hence, this needs rewriting. I would expect brief, yet concise, the quantitative description in the abstract. Response: As suggested by the Reviewer, Abstract was re-written.
- Introduction: The introduction should be improved, in order to clearly state the problem and establish a clear objective according to the title proposed for the manuscript. It is recommended to review and cite the following references: Response: Thank you for pointing this out. In the Introduction part we briefly summarized a current situation with radiowastes and focused on the role of aquatic plant in accumulation of Cs. A lack of information related to Cs accumulation by macrophytes, especially plant physiological (enzymatic) response to the metal stress has been emphasized. The list of publications suggested by Reviewer #2 is very helpful. For Introduction we used Du et al. (2020) for extending the range of aquatic plants with water hyacinth, while other sources were implemented in the the Discussion part.
- It is necessary to clearly define the objective based on the proposed title. The objective is not clear, and the novelty of the study is not evident. Response: The novelty and objectives were mentioned in the Introduction part: “The macrophytes Bacopa amplexicaulis, Elodea densa, Ceratophyllum submersum and Limnobium laevigantum were selected for the study due to reported accumulation potential of different metals in considerable quantities, whereas no data on Cs accumulation so far have been reported in these studies [20-28]. Thus, it was important to evaluate within laboratory testing, whether these plans could be used for the determination of Cs accumulation”.
- Methodology: It is necessary to mention the purity of reagents, number of analyzes, uncertainty, etc. Response: Thank you for pointing this out. The purity of reagents has been indicated. The number of analyzes was indicated for the main experiment in p.2.2., and for each assay as well.
- Why do the authors only calculate TF? Response: We agree with the Reviewer’s assessment. A brief overview of various parameters for calculating the Me accumulation in plants has been implemented in the Discussion part: “Among the TF, other parameters are widely used in aquatic toxicology, e.g., bioconcentration and bioaccumulation factors. All these factors are calculated by similar equations, while differed by considering either only dissolved phase or both dissolved and particulate phases of a contaminant in the environment [46]. Besides, root and shoot accumulation factors, as well as translocation factors are used for distinguishing the metal localization in the plant [47].
- Discussions: Development of the results and discussions is required to be in accordance with the proposed The discussion section is superficial; a more thorough discussion is required. In reviewer opinion, these sections should be rewritten. Response: The Discussion part has been amended with appropriate citations related to the proposed mechanisms of Me accumulation by plants. Thanks again for the list of references provided by the Reviewer #2.
- Conclusion: The conclusion should be rewritten based on the review of the discussions. Conclusion section is superficial. What are authors own viewpoints? What are the major findings and how they are addressing the left behind research gaps and current challenges? Response: AS suggested by the Reviewer, Conclusions part has been revised. Our conclusions consists of the main findings obtained in our study, as well as further perspectives for the use of the tested plants as bioindicators and experiments with Cs-137. One of the main original findings, i.e., the plant-specific enzymatic response to the Cs-133 has been indicated in Abstract and Conclusions.